# Determination of Fumonisins in Grains and Poultry Feedstuffs in Croatia: A 16-Year Study

**DOI:** 10.3390/toxins14070444

**Published:** 2022-06-29

**Authors:** Marijana Sokolovic, Marija Berendika, Tajana Amšel Zelenika, Borka Šimpraga, Fani Krstulović

**Affiliations:** Croatian Veterinary Institute, Poultry Centre, Heinzelova 55, 10000 Zagreb, Croatia; t_amsel-zelenika@veinst.hr (T.A.Z.); b_simpraga@veinst.hr (B.Š.); f_krstulovic@veinst.hr (F.K.)

**Keywords:** occurrence, mycotoxin, fumonisin, ELISA, HPLC, cereals, feed, poultry feed, maize

## Abstract

Fumonisins are a group of closely related mycotoxins produced by *Fusarium, Alternaria alternata* and *Aspergillus* species. Their occurrence is correlated with various factors during growth, processing and storage. Fumonisins occurrence data in the literature mainly include the B group of fumonisins (FB1 & FB2) in raw materials, showing high frequency of positive samples in a wide range of concentrations. In this study, a total of 933 grains (63.7%) and poultry feed (36.3%) samples, collected in the 16-year period (2006–2021), were analysed with commercial enzyme-linked-immunosorbent assay for detection of three fumonisins (FB1, FB2 & FB3). All positive and suspect samples were confirmed with high-performance-liquid-chromatography method with fluorescence detection. Overall, we have determined high occurrence of FBs in grains and poultry feed in all tested years, while the lowest occurrence was determined in 2019, followed by 2009 and 2008. Although, contamination levels varied from year-to-year, majority of analyzed samples in all tested years were around 1 mg/kg, while the maximum values varied from 3 mg/kg to 22.23 mg/kg. This study highlights the importance of regular monitoring of raw materials and understanding of the fate of FBs in the food chain in order to avoid undesirable health effects in animals and accompanied economic losses.

## 1. Introduction

Fumonisins, besides aflatoxin, ochratoxin, zearalenone and trichothecene mycotoxins, are of greatest concern due to their high occurrence and potential toxic effects on human and animal health. They were isolated and characterized in 1988 in South Africa after 18 years of intensive research of field outbreaks of diseases in animals. Since the first discovery, at least 30 fumonisin analogs have been characterized and classified in four main groups (A, B, C and P series). Additional groups are also suggested (i.e., D and X series, etc.). These secondary metabolites of *Fusarium verticillioides* (formerly *F. moniliforme*), *F. proliferatum*, *F. fujikuroi* and some other *Fusarium* species, *Alternaria alternata* f. sp. *Lycopersici*, *Aspergillus awamori* and *Aspergillus niger* are polyketides. Predominantly, they are produced by *F. verticillioides* and *F. proliferatum*. [1,2,3,4,5]. *Fusarium* species often cause diseases of crops in hot climates where favourable conditions such as temperature stress, insect damage and high water activities (above 0.9) occur. Most commonly they contaminate maize and maize-derived products, although other crops can also be affected. It has been estimated that B-type fumonisins (FB1, FB2, FB3 and FB4) are the most common natural FBs in feed (more than 95% of the all detected FBs). The optimal conditions for FBs production are at high temperature (30 °C) with water activity (aw) of 0.98 and 400 ppm of CO_2_ [6,7] FBs are a long-chain aminopolyols with tricarballylic acid side chains that differ in the number and position of hydroxyl groups at the backbone. They are considered heat stable (in common food processing), but a number of different forms have been described as a result of hydrolysis (hydrolysed fumonisins B and partially hydrolysed fumonisins B), formation of Maillard-type modified forms (NCM- and NDF-fumonisins B), and interaction with the matrix (hidden and bound forms) [8,9,10].

It is well known that acute toxicity of FBs is low, but high concentrations of FBs can cause neurotoxicity, nephrotoxicity, hepatotoxicity, immunotoxicity, and can elicit toxic effect in alimentary system in various animal species and humans. In animals, FBs are correlated with equine leukoencephalomalacia, porcine pulmonary oedema, liver toxicity and liver tumour promotion in rats and other species and are implicated in the onset of certain diseases in humans [11]. According to the International Agency for Research on Cancer (IARC), FB1 derived from *F. verticillioides* is evaluated as Group 2B, i.e., a possibly carcinogenic to humans, due to the lack of evidence for the carcinogenicity of FBs in humans but sufficient evidence in experimental animals [12].

Knowledge on adverse health effect of B-type FBs in avian species show low oral absorption, rapid plasma elimination and hepatic half-elimination time of several days. The 50% lethal dose for FB1 on chicken embryo is 18.7 g per egg (route of inoculation via airspace) [13]. Presence of these toxins in feed, in the quantities around the European Union (EU) guidance level, have been correlated with residues in the liver, hepatic oxidative stress, impaired immune system and intestinal health, increased sphinganinine (Sa) and Sa/sphingosine (So) levels (at concentration 2 mg FB1/kg feed), increased activity of aspartate aminotransferase (AST) (marker of organ damage and toxicosis), lower feed intake, maldigestion of vitamin D, calcium or phosphorus, secondary lower body weight gain and rickets [14]. Due to similar toxicological profiles and potencies of B-type fumonisins, it is considered that they have similar mode of action at the cellular level (disruption of sphingoid metabolism and toxicity) and DNA damaging effect. Considering the aforementioned half-elimination time, a certain quantity of FBs can accumulate in the body. Namely, in the research of Tardieu et al. [15], the highest cumulative concentrations of FBs were detected in muscle and liver of chickens and turkey when they were feeding with contaminated feed (contamination levels were: 0.06 mg/kg and 0.10 mg/kg). According to current knowledge, a No Observed Adverse Effect Levels (NOAELs) of 8, 20 and 20 mg/kg feed (2 mg/kg body weight per day) were identified for ducks, chicken and turkey, respectively [10,16,17].

For FBs (FB1 + FB2) current European legislative provide recommendation (2016/1881/EC). Guidance values, in mg/kg relative to feedstuffs with a moisture content of 12%, are 60 for maize and maize by-products and 20 for poultry feed. Irrespective of the fact that feed ingredients or compound feed might be considered acceptable according to this recommendation, special attention should be given when animals are feeding with contaminated feedstuffs. It is important to evaluate the FBs contamination level and the quantity that will be given to animals to avoid exposure of animals to high levels of these mycotoxins.

There are several methods for detection of FBs in feed including enzyme-linked immunosorbent assay (ELISA), different immunoassays, capillary electrophoresis, thin layer chromatography, and high performance liquid chromatography methods coupled with different detectors. Specificity of determination of FBs is the lack of UV-absorbing or fluorescent chromophores. Application of high performance liquid chromatography (HPLC) with fluorescence detection (FLD) or diode array detection (DAD) can be done after derivatization with fluorescent labels [10,18,19,20,21]. Coupling of HPLC with tandem mass spectrometry (LC-MS) offers easier sample preparation and greater sensitivity. However, poor recovery and accuracy has been documented in application of some multi-toxin methods (caused by different polarity of toxins and matrix effect) [10]. Therefore, determination of B group of FBs with screening assays and confirmation with validated chromatographic method is still used in a great number of laboratories worldwide.

A number of surveys on fumonisin occurrence in feed in Croatia and other European countries conducted by different research groups, showed frequent contaminations at various contamination levels. Occurrence studies on FBs are mainly focused on the presence of FB1 and FB2 in raw materials and only small scale of studies included FB3 and their presence in poultry feed [10,21,22,23,24,25,26,27,28,29,30,31,32].

The aim of this research was to evaluate the occurrence of FBs (FB1 + FB2 and FB3) in maize samples and poultry feed during a sixteen-year period. For that purpose, we have analysed samples with the commercial enzyme-linked immunosorbent assay (ELISA) and confirmed all suspect and positive samples with high-performance liquid chromatography method with fluorescence detection (HPLC-FLD).

## 2. Results 

### 2.1. Occurrence of Fumonisins in Grains

The highest occurrence of FBs in grains (over 90% of analyzed samples) was in the years 2012–2016, 2018 and 2021 and lowest in the years 2008 and 2019 (Figure 1). 

### 2.2. Occurrence of Fumonisins in Poultry Feed

In poultry feed samples, highest occurrence (over 90% of analysed samples) was detected in the years 2006, 2011, 2013–2016, 2018, 2019 and 2021 (Figure 2). 

### 2.3. Occurrence of Fumonisins in Grains and Poultry Feed

Overall, we have determined high occurrence of FBs in grains and poultry feed in all tested years, while the lowest occurrence was determined in 2019, followed by 2009 and 2008 (Figure 3). 

Data on occurrence of total FBs in grains and feed are presented in Table 1. 

## 3. Discussion

Moulds and their toxins are common contaminants of a wide range of raw materials and compound feed worldwide. Incidence of *Fusarium* species is considered as rather frequent in cereals in all European countries, especially in maize [29,33,34]. In the survey on mycotoxin occurrence in feed, FBs (FB1, FB2 and FB3) were determined in 0%, 33% and 70% of the analysed samples from Northern, Eastern and South Europe, respectively [10,22,23]. The survey covered a period from January 2004 to December 2011 and included samples from Asia, Europe, Oceania, America, Africa and the Middle East. Only 4% of samples exceeded the applicable European guidance value. In all tested years, more than 60% of maize and compound feed samples (for swine, poultry and dairy cow) were positive on FBs. In a more recent survey in maize grains in Spain, FBs were the only detected mycotoxins during harvest and storage. The authors also noticed increased levels of FBs during harvest for all analysed years (2016, 2017 and 2018), but only in 2018 year they documented FBs after three months of storage. Contamination with FB1 in the pre-harvest maize samples was in the range from 0.20 mg/kg to 5.90 mg/kg and for FB2 from 0.09 mg/kg to 0.74 mg/kg. In stored maize samples, FB1 concentrations were in the range from 0.18 mg/kg to 0.20 mg/kg, while FB2 was not detected [24]. A great number of other authors have also documented high occurrence of FBs in various commodities. Indicating constant threat of these toxins [10,23,25].

There are several research publications on occurrence of FBs in Croatia. They became focus of research since FBs were considered as aetiological agents of nephrotoxic conditions in humans. In the research of Peraica et al. [26], frequency of FB1-positive samples in maize was rather high and in correlation with the slightly higher values of the temperature and rainfall in this geographic area. In 2002, analysis of maize grain samples showed incidence of FB1 of 100%, while FB2 was present only in 6% of analysed samples. Mean concentration of FB1 was 0.46 mg/kg and the range was from 0.14 mg/kg to 1.38 mg/kg. FB2 concentrations in three positive samples were 0.07 mg/kg, 0.11 mg/kg and 3.08 mg/kg. No data on FB3 in this study were reported [27]. In a research of Cvetnić et al. [28] on the frequencies and distribution and the toxigenic potential of *Fusarium* species, isolated from non-harvested maize left in the field over winter in 1999 and 2003 in northern Croatia, they found incidence of *Fusarium* species of 78.6% and 85.0% in 1999 and 2003, respectively. *F. verticillioides* was a dominant species found in 12.5% (1999) and 35.7% (2003) of maize samples. Same authors documented production capability of FB1 in all strains isolated in 1999 (in concentration range from 280 to 918 mg/L) and in 55% of analysed strains isolated in 2003 (in the concentration range from 48 to 400 mg/L). In the period from 2002 to 2008, Ivić et al. [29] analysed samples of wheat, maize, soybean and pea for the presence of *Fusarium* species and found contamination levels of 5–69%, 25–100%, 4–17% and 3–17%, respectively. *F. verticillioides* was again the dominant species on maize (83% isolates). High occurrence (range from 67.4% to 100%) of FBs in grains and animal feed in 2012 was described in several publications by Pleadin et al. [30,31,32]. Due to the lack of continuous data on incidence of FBs and the fact that the majority of our maize sample originate from this area, it is reasonable to assume that there is the strong potential of high incidence of FBs in grains and poultry feed in Croatia.

In this study, we have documented year-to-year variations in total FBs content in grains and feed collected from Croatian storage facilities (Table 1). It is well known that such variation are often correlated with the incidence of fungal diseases in the field, like the one caused in maize by *F. verticillioide, F. proliferatums* and *F. graminearum*. While *F. verticillioides* grows better in the dry years and during the late maturity stage, *F. graminearum* is more common during dry season [1,35]. Aforementioned highlights the importance of monitoring climate conditions in respect to incidence of *Fusarium* species and their mycotoxins.

Croatia is a country that belongs to the Adriatic-Mediterranean and Pannonia-Danube group of countries situated in the Central Europe. The storage facilities that were included in this study are situated in the lowland region that according to Köppen-Geiger climate classification system belongs to the climate type C (temperate). The lowland of northern Croatia represent 53% of area of the country with the mean temperature from 10 °C to 12 °C and wetter precipitation conditions. Due to the climate change, usual climate patterns (warm summers and cold winters and more precipitation in spring and autumn) are changing. Observed are warming (warmer daytime temperatures and warmer nights) and the increase of maximum temperatures of 0.3 °C to 0.4 °C per decade, while the precipitation trends have decreased most significantly in the summer (Appendix A). For both parameters, certain inter-annual variations exist [36]. A number of predictions of influence of climate change on fungal colonisation and mycotoxins productions have already been made [37,38]. The trend of increasing temperatures, with spike from April to June and again in September and October will result in the heat and extreme heat conditions that might significantly create problems in the agricultural sector [36]. It has been shown that *F. verticillioides* can easily adapt to various climate change conditions and continue to grow on crops and produce FBs [7]. Figures in Appendix A show temperature and precipitation trends that might be correlated with high in-coincidence of FBs in samples in aforementioned years. This highlights the importance of understanding the trends and incidence of moulds and mycotoxins and factors that favour their growth in order to implement adequate preventive measures for better control and management of FBs and other mycotoxins in grains and compound feed. Considering the crop production in Croatia, production statistics (2006–2019) indicate that Croatia represent up to 3.17% of production of maize in the European Union [7]. Although there are around 1.3 million hectares (ha) of agricultural land, and the country has a potential of being self-sufficient for the production of maize, agricultural products are still imported at the exchange rate for agriculture and related products around 6.6 [39]. It is reasonable to assume that a part of the grains available on the market is not of Croatian origin. Furthermore, the samples collected from the storage facilities were acquired from different sources. Therefore, we were not able to collect information on other important factors such as: plant genetics (resistant or not), environmental conditions in growing area, temperature (maximum daytime temperatures, minimum night-time temperatures, precipitation, atmospheric CO_2_ levels, humidity, soil nutrient and moisture content, pH, insect damages, presence of other microorganisms, agricultural practices, transport and storage conditions. Therefore, the impact of climate change and environmental conditions in this study cannot be directly correlated with the incidence of FBs, although certain trends might be visible (years with high temperature and low precipitation will probably have higher FBs contamination in the field). According to Scudamore and Patel [40], distribution of FBs is more even than other toxins like deoxynivalenol, indicating that results on incidence of FBs might reflect the actual general level. The results of this research show, for the first time, the long-term occurrence of FBs (FB1 + FB2 + FB3) in grains and poultry feed in Croatia, indicating the potential economic and health risk in this country.

In respect to contamination levels in our study, in the period from 2006 to 2009, the majority of tested samples were below 1 mg/kg of total FBs, with the maximum levels were below 3 mg/kg for both grains and feed samples. In a research of Griessler et al. [41] on occurrence of mycotoxins in Southern Europe (Greece, Cyprus, Spain, Italy and Portugal) in different feed and raw materials collected from January 2005 until August 2009, the maize was the most affected raw material and FBs were present in 68% of samples analysed with HPLC, with an average value of 2.20 mg/kg (median: 1.67 mg/kg, range 0.09–7.71 mg/kg) and 6.31 mg/kg in samples analysed by ELISA. In finished feed samples (pigs, horses, rabbits and pet animals) and other commodities, occurrence of FBs was 40% (mean: 0.62 mg/kg, median 1.37 mg/kg, range 0.10–3.23 mg/kg) and 100% (mean: 0.75, median 0.11, range: 0.03–3.09 mg/kg), respectively. Higher values were described for maize samples tested with ELISA method. Comparing our results for this time period, we have detected similar incidence of FBs but a lower level of contamination. Similar results with high occurrence of FBs (FB1 + FB2 + FB3) in maize originated from France (maximum concentrations 5.0 mg/kg) and Argentina (maximum concentrations 10.0 mg/kg) in the period from 2004 to 2007 were reported by Scudamore and Patel [40]. In Italy, Berardo et al. [42] analysed 2258 grain samples over a three-year period (2006–2008) and noticed high FBs (FB1 + FB2 + FB3) occurrence in maize samples in 2006 (maximum value 10.9 mg/kg) and lower in 2008 (maximum value 4.8 mg/kg). These year-to-year variations were evaluated on different geographical areas and correlated with specific environmental conditions during the growing and harvest season. Our results are in agreement with these data, since we have also detected lower mean and median values in 2008 in respect to 2006 year. There is only a limited number of studies on FBs in grains (mostly maize) and feed (mostly non poultry feed) in Croatia. By evaluation of available published data, our overall results for 2007 are not in correlation with the data published by Klarić et al. [43] that found overall incidence of FBs of 27% with mean concentration of 3.69 mg/kg. Detected overall incidence of 69.2% and median value of 0.94 mg/kg in our study might be explained with different types of samples and/or their origin.

In the period from 2010 to 2013, the median concentrations of FBs were still below 1 mg/kg, but we noticed slight increase in maximum levels of FBs in grains for the years 2010 and 2012 (5.85 mg/kg and 7.31 mg/kg respectively). Our overall results (Figure 3) are in correlation with the data on occurrence of mycotoxins in the study of Schatzmayr and Streit [22] for the period from 2004 to 2012 that applied the same methods (ELISA and HPLC) on 19757 samples of feed and feed raw materials worldwide. In Croatia, there were several studies on incidence of FBs in 2012 showing occurrence of FBs in maize samples in Croatia with 67.4% (range 0.03–25.2 mg/kg) and 100% (range 0.03–1.15 mg/kg). Our data are in concordance with the 100% occurrence but with different maximum (7.31 mg/kg) contamination levels of FBs (our median contamination level was 0.56 mg/kg). In a neighbouring country Serbia, Kos et al. [44] have analyzed 204 maize samples in the period 2012–2015. The highest median concentrations were detected in 2012, 2013 and 2014 (4.44 mg/kg, 4.25 mg/kg and 4.86 mg/kg) in respect to the year 2015 (2.23 mg/kg). The authors have analyzed additional FBs including FB4, A1 and A2. While FB4 were detected in the samples in quantities comparable to FB3, fumonisin A1 and A2 were detected at levels below 0.02 mg/kg [22]. In comparison with these data, median and maximum values of maize samples in our study were lower for all tested years, and only maximum value in the year 2015 was in correlation with the results in the study from Serbia. In Table 2, we have summarized the values for FB1, FB2 and FB3 from research on FBs occurrence in Croatia and in several countries in Europe. For the purpose of comparison with our data, and set guidance levels in EU, the data are presented in mg/kg units (irrespective of the units in original publication. It is clear from the data that FBs have probably been present in different commodities in Croatia and in other countries in this climatic region, but the onset of awareness of their potential harmful effects and availability of more sensitive analytical methods helped in defining the risk of these mycotoxins. However, well-known climatic changes are probably the most important contributing factor for the incidence of higher maximum concentration levels noticed in the more recent surveys.

In 2014 and 2015, we have detected higher contamination levels, indicating similar trend. Kos et al. [44] have evaluated their data in relation to environmental conditions (2012, 2013 and 2015 were hot and dry years, while 2014 was extremely wet) and the presence of *F. verticillioides*. Noticed differences highlight the importance of evaluation of a great number of different factors, if these variations need to be explained.

In the following three years (2014–2016), the majority of grains contained FBs values above 1 mg/kg with the higher maximum levels in grains in 2014 and 2015 (9.30 mg/kg and 6.42 mg/kg) and in feed in 2016 (5.72 mg/kg). High occurrence of FBs was also detected in the global mycotoxin survey of feed (2008–2017) reported by Gruber-Dorninger et al. [25]. FBs were detected in 60% of grains and feed samples with the median value of 0.72 mg/kg and the extremely high maximum values of 290.52 mg/kg. In the same study, 73% of finished feed (median value of 0.56 mg/kg and maximum value of 290.52 mg/kg), 80% of maize (median value 1.3 mg/kg, maximum value 218.88 mg/kg) and only 14% of wheat samples (median value 0.14 mg/kg, maximum value 28.28 mg/kg) were contaminated with FBs (sum of B1 + B2 + B3). The reason for these high values might be explained by various contributing factors such as susceptibility of grains to fumonisin contamination, pre- and post-harvest agricultural practices, differences in climatic conditions during plant growth (high temperatures and low precipitation around silking) and conditions during transport and storage. This study included Croatia in the geographic area “Southern Europe”, where the incidence of FBs was 74.9% with the median value of 0.64 mg/kg. The year-to-year variation in this study shows no data on FBs for the years 2008 and 2009. In the feed samples, FBs were detected in the concentrations around 1.5 mg/kg in 2010, declined to 0.5 mg/kg in 2011 and the peak was in 2013 (contamination level around 1 mg/kg). The concentration of FBs in tested feed samples declined again until 2016 (concentration level around 0.5 mg/kg) and increased in 2017 (concentration level around 0.8 mg/kg). In maize, median concentration of FBs in 2010 were around 1.5 mg/kg, increased until 2013 to 2.6 mg/kg, decreased in 2014 to 1.8 mg/kg, increased in the following two years up to 2.5 mg/kg and in 2017 were around 2.1 mg/kg. The BIOMIN Mycotoxin Survey project started in 2004 with the aim to provide the most comprehensive data set on mycotoxin occurrence in the agricultural commodities used for production of feed. They have analysed a great number of samples worldwide for the presence of six mycotoxins (aflatoxin B1, fumonisin, zearalenone, deoxynivalenol, ochratoxin A and T-2 toxin). Besides valuable data on worldwide occurrence of these mycotoxins, the data were also evaluated in respect to different contributing factors (including climate conditions) to provide information of the most important contributing factors for mycotoxin production and identification of the potential risk of mycotoxin on animal health [51]. In comparison to BIOMIN global study results for our region, we have noticed increase at similar contamination level (median) for FBs in maize from 2012 to 2014, decrease until 2016 and again increase in 2017 year. In poultry feed samples, increase in median concentrations of FBs was detected from 2010 to 2012 and from 2013 to 2016, while in 2017 we noticed lower median concentrations in relation to previous years. This slight discrepancy in years can be explained with the calculation of analyzed years. Namely, they have defined a year according to approximate seasons of crop growth and harvest, so the samples were collected for this region from October to the end of September of the subsequent calendar year. Additionally, samples of feed included samples of feed for other animals, while in our study we have only analyzed poultry feed. Comparing the results of the occurrence study in neighbouring country, Bosnia and Herzegovina [52], our data are in correlation in respect to variation in the years 2013–2015 (same peak in 2014 year for maize samples). However, we have detected higher median and maximum concentrations in our maize samples in the years 2014 and 2015.

In 2017 and 2018, when the FBs content was found to be significantly higher at the global level [25], detected FBs median concentrations in grains were 0.82 mg/kg (maximum 2.84 mg/kg) and 0.20 mg/kg (maximum 4.15 mg/kg), respectively. Contamination of feed in these two years was even lower. In the period from 2015 to 2019, Tarazona et al. [50] analyzed 98 samples of maize kernels and found that FBs (FB1 + FB2) were the most common mycotoxins found in maize and that certain number of samples (20.4%) were not in compliance with the EU limit for maize. They have recorded the highest concentrations in the years 2016 and 2017 in comparison to years 2015 and 2019. Described variations are not in accordance with our result that might easily be explained with environmental conditions during growth of maize in the field. Similar results in this area of Europe (for previous years) were found by other authors that also documented increased occurrence of FBs during storage [24,53]. These findings might also explain higher contamination levels of feed samples in the seven studied years in our study that was significantly evident in 2016. In 2018, European Food Safety Authority (EFSA) has published report on Risks for animal health related to the presence of FBs [10]. The report includes analysis of 7970 samples of feed collected in the period between 2003 and 2016 from 19 European countries. The majority of countries reported results on FB1 and FB2, and only three countries reported data that included FB3. High FBs concentrations were reported for maize (range 0.02–2.04 mg/kg), wheat (range 0.0004–2.48 mg/kg), as well as in samples of complementary/complete feed (range 0.0003–1.68 mg/kg). Data also included poultry feed: feed for fattening chickens, ducks and turkeys, laying hens and poultry starter diets. Mean and median values of FBs (FB1 + FB2 + FB3) for fattening chickens (11 samples) were 0.10–0.49 mg/kg and 0–0.28 mg/kg, respectively. For poultry starter diets (175 analysed samples), mean and median values were 0.40–0.47 mg/kg and 0.13–0.18 mg/kg. For laying hens (18 samples), mean and median values were 0.35–0.68 mg/kg and 0.03–0.30 mg/kg. For fattening turkeys/complete feed (two samples), mean and median values were 0.50–0.61 mg/kg and 0.46–0.61 mg/kg. For ducks (nine samples) the values were 0.60–0.64 mg/kg and 0.24–0.30 mg/kg. In comparison to these results, incidence of FBs in poultry feed samples were higher in all tested years. Based on the data on incidence of FBs in their survey, EFSA has made estimates of exposure to FBs in poultry, showing that diet concentration of FBs (expressed on the dry feed matter) calculated as mean (95th percentile) for fattening chickens is from 0.37 (1.52) mg/kg to 0.60 (1.75) mg/kg. For laying hens the mean (95th percentile) diet concentration were from 0.33 (1.40) mg/kg to 0.56 (1.57) mg/kg. For fattening turkeys the mean (95th percentile) diet concentration were from 0.058 (0.072) mg/kg to 0.27 (0.38) mg/kg. For fattening ducks the mean (95th percentile) diet concentration were from 0.078 (0.079) mg/kg to 0.31 (0.45) mg/kg. If the samples contained higher amounts of FBs in feed, such as those detected in our study, the exposure to these toxins might be even higher and thus present a potential significant risk for animal health.

In the last three years of this study (2019–2021) we have noticed an increase in the maximum contamination levels (22.23 mg/kg in grains), although only samples in 2020 included majority of samples above 1.23 mg/kg in feed and grains. In a recent survey of Raj et al. [54] of corn samples that originated from different regions of Croatia and neighbouring countries Serbia and Bosnia and Herzegovina in the year 2021, they found high incidence of FBs (FB1—79%, FB2—73%, no data on FB3). The median (and maximum) concentration levels of FB1 and FB2 were 2.88 mg/kg (28.75 mg/kg) and 0.82 mg/kg (15.48 mg/kg), respectively. In 2021, we have found even higher incidence of FBs in maize (90%), similar maximum concentration level (22.23 mg/kg), but lower median concentration values of 0.5 mg/kg.

In summary, mycotoxins have gained a lot of attention in the recent decades that resulted in a great number of published data on occurrence of mycotoxins in different commodities. Since, there are great discrepancies in relation to origin and type of samples, methodologies and the sampling years, realistic evaluation of the occurrence of FBs is a challenging task. However, research on incidence that provides data like the ones gathered in this research are important because they can help understand the FBs occurrence and contamination levels and their trends in the selected geographic area. This data might be useful for feed industry and poultry producer for mitigating harmful effects of mycotoxins and selection of adequate preventive measures.

In addition, since the majority of analysed samples of grains and poultry feed in this survey were below the EU guidance values set by the Croatian and EU regulations (EC/1881/2006), these samples were used as a feedstuff for animals. Furthermore, certain fungal species might in addition to FBs, produce other mycotoxins [55,56], making the final “multi-toxin concentration level” in compound feed pretty challenging for animal health, especially after a long-term exposure.

## 4. Conclusions

Contamination of grains and feed with *Fusarium* species is a significant mycotoxicological risk due to the often contamination of these moulds and their toxigenic potential. 

In our study, we have applied optimized and validated analytical methods (ELISA and HPLC-FLD) for the quantification of three B-group fumonisins (FB1, FB2 & FB3) in 933 samples (maize and wheat grains, soybean seed and compound poultry feed) in the 16-year period (2006–2021). The samples originated from different facilities settled in the lowland of northern part of Croatia.The results of this survey indicate that FBs occur frequently in grains and poultry feed marketed in this part of Croatia, and that FBs require the attention of the feed industry and poultry producers. The lowest occurrence of FBs in grains and poultry feed was determined in 2019, followed by 2009 and 2008.Contamination levels of the majority of analyzed samples in all tested years were around 1 mg/kg, while the maximum values varied from 3 mg/kg to 22.23 mg/kg. Detected variations of incidence and contamination levels of FBs in analysed samples were probably associated with different contributing factors that favour growth and toxigenicity of *Fusarium* moulds during plants growth and/or storage.Such high occurrence of FBs (even at concentrations below the maximum tolerable levels) might lead to long term exposure of animals to FBs leading to undesirable health effects and significant economic losses. For the Risk Assessment of potential harmful effects of FBs on animal health, it is important to include all the data about production, transport and storage of raw materials and compound feed.Since environmental factors are important for infection of grains with *Fusarium* species and production of FBs, prevention of mycotoxin contamination in the field and later during transport and storage is the necessary step in management and control of contamination of raw materials and compound feed. Constant monitoring using available rapid and accurate methods in grains and feed will aid in compliance with current regulations and production of safe high quality feed.

## 5. Materials and Methods

### 5.1. Materials and Methods

A total of 933 samples of grains (maize grains, wheat grains and soybean seeds) and poultry feed (feed for broilers, laying hens and uncategorized poultry feed) were collected during the period from 2006–2021 from several storage facilities situated in the lowland of northern Croatia (Appendix A, Appendix A). Croatia is situated in southeast Europe adjoining the Mediterranean, central and southeast Europe. It lies between latitude 42°23′ and 46°33′ north, and between longitude 13°30′ and 19°27′ east. Geographical coordinates of this sampling area are following north (46°33′ N, 16°22′ E), west (45°28′ N, 14°54′ E), south (44°50′ N, 15°21′ E), and east (45°44′ N, 18°26′ E). All samples were taken regularly at the time of filling and in the presence of authorized doctor of veterinary medicine. The samples derived from feed factories that included monitoring of mycotoxins in their control and management programs. The only available data on raw material origin was that it was marketed in this part of Croatia. Since sampling plays one of the most important parts in the precision of the determination of the levels of mycotoxins, it has been done in accordance with the Commission Regulation of European Union (EC/401/2006). To achieve the representative laboratory sample, the primary samples were homogenized and quartered to obtain a 1 kg laboratory sample. All samples were stored in a refrigerator (2–8 °C) and adequately ground for analysis using ultra centrifugal mill ZM200 (Retsch^®^, Hann, Germany).

### 5.2. Standard Solutions

Commercial Enzyme Linked Immunosorbent Assay (ELISA) was purchased from the producer Romer Labs Diagnostic GmbH, Tulln, Austria. FBs standards were purchased from Sigma (St. Louis, MO, USA). All chemicals were purchased from Merck (Darmstadt, Germany). Mycotoxin standard solutions for spiking or HPLC analysis were prepared in methanol by appropriate dilution of the stock standard solution. Standard solutions were stored at −20 °C. In this study, we did not analyse fumonisin B4, since it occurs mainly in grapes and is not evaluated as toxic due to the lack of data on toxicity and toxicokinetics [10]. According to available data in literature, modified forms occur in rather low numbers (<10%) and were also not included in this study.

### 5.3. Analytical Methods 

All samples were analyzed with commercial direct competitive enzyme-linked immunosorbent assay (ELISA) AgraQuant^®^ Total fumonisin Assay 0.25/5.0 Romer Labs^®^ (Romer Labs Diagnostic GmbH, Tulln, Austria). Suspect and positive samples were confirmed by High performance liquid chromatography method with fluorescence detection (HPLC-FLD) described by Sydenham et al. [57] with slight modifications. Both methods were validated for determination of FBs (FB1, FB2 & FB3) in maize, wheat, soybean and compound poultry feed to assure compliance with the guidance and performance criteria set in European Regulations EC/2006/401, EC/2004/882 and EC/2017/625. For each matrix we have evaluated following methods performance criteria: limit of detection (LOD), limit of quantification (LOQ), linearity, trueness (expressed as average value of 20 repetitions in repeatability conditions for the selected concentration levels), precision (expressed as relative standard deviation value of 20 repetitions for repeatability (RSD_r_) and reproducibility(RSD_R_)), specificity (percentage of positive samples in positive spiked sample and percentage of negative samples in analysis of blank sample). 

Twenty grams of each homogenized ground sample was used for both analyses. Spiked samples were prepared according to the described methodology for FBs. If the results showed higher concentrations than the standards used in the analysis, quantity was determined after second analysis using different spike sample and/or standards.

#### 5.3.1. Enzyme-Linked Immunosorbent Assay (ELISA)

Direct competitive enzyme-linked immunosorbent assay (ELISA) AgraQuant^®^ Total fumonisin Assay 0.25/5.0 Romer Labs^®^ (Romer Labs Diagnostic GmbH, Tulln, Austria) has been used according to manufacturer instructions. In short, twenty grams of sample was extracted with methanol/water (70/30, *v*/*v*, 100 mL) by shaking for three minutes. After five minutes, the top layer was filtered through a cellulose filter (particle retention of 11 µm and ash content ≤ 0.06%) and the sample extract was further diluted in ration 1:20 with de-ionized water. All the results that were not in the range of used standard solutions (0.25–5.0 mg/kg) were diluted and reanalyzed. Optical densities (OD) of samples and standard in the assay were measured with an absorbance filter of 450 nm and differential filter of 630 nm. The results were calculated from the dose-response curve of five standards (0, 0.25, 0.5, 1.0 and 5.0 mg/kg) using Log/Logit regression model. The results were considered accurate if the linearity coefficient of the calibration curve was ≥0.990 and OD values for the zero standards were higher than 0.5 absorbance units. Limit of detection (LOD) was calculated from the results of the average values of ten FBs free samples plus values of three standard deviations and compared with the LOD provided by the manufacturer (i.e., 0.20 mg/kg). Limit of quantification (LOQ) was calculated from the results of the average values of ten FBs free samples plus values of ten standard deviations and compared with the LOQ provided by the manufacturer (i.e., 0.25 mg/kg).

#### 5.3.2. High Performance Liquid Chromatography Method with Fluorescence Detection (HPLC-FLD)

All suspect and positive samples were confirmed with the modified HPLC method with post column derivatisation and fluorescence detection (HPLD- described by Sydenham et al. [57]. The method has been adopted, optimized and validated before routine use according to aforementioned method performance criteria. In short, twenty grams of samples were extracted with methanol-water (3:1, *v*/*v*, 100 mL) for three minutes, centrifuged, and aliquot was used for clean-up step on Extract CleanTM C-18 columns (Grace Davison Discovery Sciences, Deerfield, IL, USA). Derivatisation has been done with o-phthalaldehyde (OPA). Chromatographic analysis was performed on the HPLC Agilent 1100 series (Agilent, Waldbronn, Germany) equipped with a quaternary pump, autosampler, column thermostat, diode array detector and fluorescence detector. Quantification has been made on C-18 column by measurement of the peak areas of FB1, FB2 and FB3 (retention times using selected separation protocol and solvents were: 7, 21 and 24 min respectively) and comparing with the calibration curve in Agilent ChemStation^®^ software (OpenLAB CDS, Chem Station Edition for LC&LC/MS Systems, Rev.A.01.03.111). The flow rate of the mobile phase (methanol/0.1 M sodium phosphate, 80/20, *v*/*v* at pH 3.3) was 1 mL/min. Limit of detection (LOD) was calculated as three-times the signal-to-noise ration and limit of quantification (LOQ) was calculated as ten times the signal-to-noise ration. All samples were tested in duplicates. 

### 5.4. Statistical Analysis 

For both methods, all samples were analyzed in duplicates, in series of eight to ten samples using additional spiked sample for recovery test. The results of the contaminated test were corrected for the recovery of the spiked sample for each series. The data were analysed by the Excel Data Analysis ToolPak (Microsoft Office Excell 2007). The recovery of both methods was calculated by dividing the concentration value obtained for the spiked feed samples with concentration spiked and multiplied by 100 to express it in percentage. 

The results of evaluated performance criteria for each tested matrix indicated that the methods are fit-for-purpose. In short, the LOQ were 0.25 mg/kg for ELISA test and 0.01 mg/kg for HPLC-FLD method. Linearity was evaluated with each testing. The results were considered accurate if the linearity coefficient of the calibration curve was ≥0.990 for both methods. SD values of evaluation of the parameter Trueness for in all tests (different spiking levels and different matrices) were below 30.00. Precision parameters in repeatability (RSD_r_) and reproducibility conditions (RSD_R_) were below ≤30% for contamination level ≤0.5 mg/kg and below 20% at the contamination level >0.5 mg/kg. Precision parameters in reproducibility conditions (RSD_R_) were below ≤60% for contamination level ≤0.5 mg/kg and below 30% at the contamination level >0.5 mg/kg. Specificity for spiked and blank samples in all tests was 100%. Testing of samples in duplicates showed that results were reproducible and that tested FBs were identifiable and accurate. In HPLC-FLD, method there for no interfering peaks at LOQ for each of the FBs in each tested matrix. The mean recoveries for total (FB1 + FB2 + FB3) were 93% for ELISA tests and 97% for HPLC-FLD method. Mean coefficients of variation were 10.5 for ELISA and 8.4% for HPLC-FLD method.

## Figures and Tables

**Figure 1 toxins-14-00444-f001:**
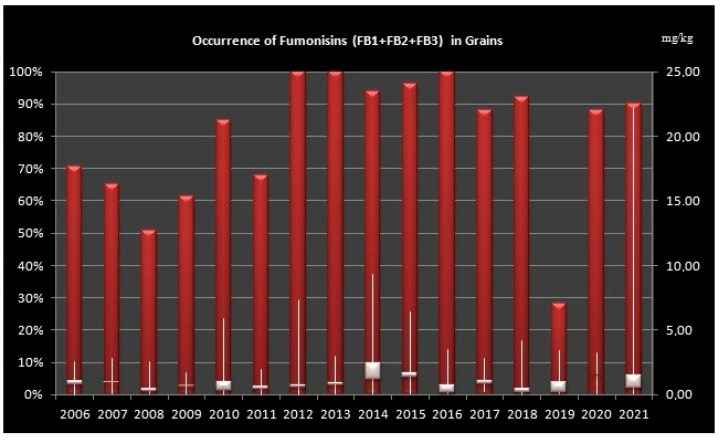
Levels of Fumonisins (B1, B2 and B3) in grain samples from Croatia during sixteen-year period (2006–2021). Left Axis: percentage of positive samples, Right Axis: concentration of fumonisins in mg/kg, Bars shows minimum, maximum and median values.

**Figure 2 toxins-14-00444-f002:**
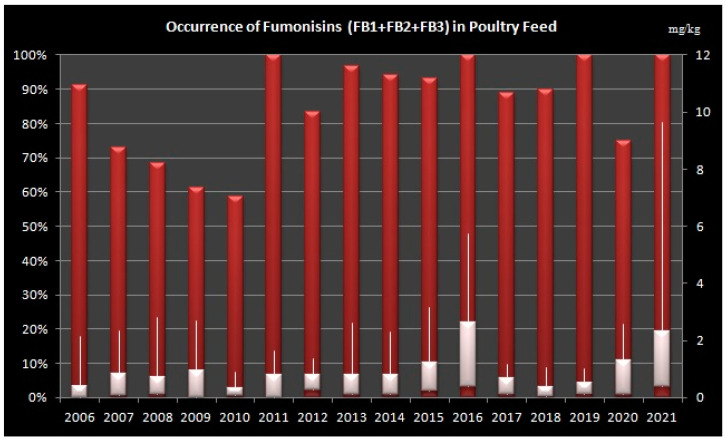
Levels of total Fumonisins (B1, B2 and B3) in Poultry Feed Samples from Croatia during sixteen-year period (2006–2021). Left Axis: percentage of positive samples, Right Axis: concentration of fumonisins in mg/kg, Bars shows minimum, maximum and median values.

**Figure 3 toxins-14-00444-f003:**
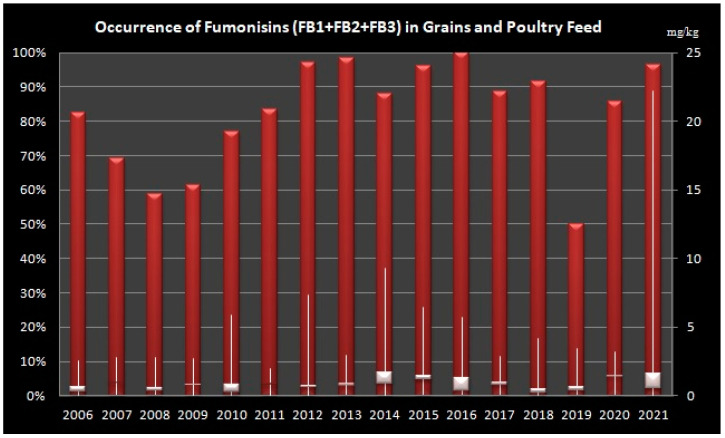
Levels of Fumonisins (B1, B2 and B3) in Grains and Poultry Feed Samples from Croatia during sixteen-year period (2006–2021). Left Axis: percentage of positive samples, Right Axis: concentration of fumonisins in mg/kg, Bars shows minimum, maximum and median values.

**Table 1 toxins-14-00444-t001:** Occurrence of Fumonisin (B1 + B2 + B3) in Grains and Poultry Feeds in Croatia during the sixteen-year-period (2006–2021).

Year	Samples	No. of Samples	FB1 + B2 + FB3 (mg/kg)		
		Positive/Total (%)	Mean	Range	Median
2006	Grains	12/17 (70.6)	1.07	0.01–2.52	0.77
	Feed	21/23 (91.3)	0.42	0.01–2.14	0.10
	Total	33/40 (82.5)	0.67	0.01–2.52	0.29
2007	Grains	28/43 (65.1)	1.02	0.10–2.81	0.98
	Feed	35/48 (72.9)	0.86	0.09–2.33	0.84
	Total	63/91 (69.2)	0.93	0.09–2.81	0.94
2008	Grains	33/65 (50.8)	0.48	0.10–2.57	0.24
	Feed	37/54 (68.5)	0.72	0.10–2.79	0.44
	Total	70/119 (58.8)	0.60	0.10–2.79	0.35
2009	Grains	24/39 (61.5)	0.70	0.10–1.67	0.66
	Feed	19/31 (61.3)	0.94	0.01–2.68	0.86
	Total	43/70 (61.4)	0.81	0.01–2.68	0.76
2010	Grains	57/67 (85.1)	1.00	0.01–5.85	0.30
	Feed	17/29 (58.6)	0.34	0.06–0.89	0.15
	Total	74/96 (77.1)	0.85	0.01–5.85	0.26
2011	Grains	19/28 (67.9)	0.66	0.01–1.95	0.43
	Feed	26/26 (100)	0.81	0.01–1.63	0.92
	Total	45/54 (83.3)	0.75	0.01–1.95	0.87
2012	Grains	55/55 (100)	0.75	0.01–7.31	0.56
	Feed	10/12 (83.3)	0.81	0.26–1.37	0.88
	Total	65/67 (97.0)	0.73	0.01–7.31	0.57
2013	Grains	30/30 (100)	0.93	0.07–2.95	0.73
	Feed	29/30 (96.7)	0.80	0.10–2.60	0.71
	Total	59/60 (98.3)	0.87	0.07–2.95	0.72
2014	Grains	24/27 (88.9)	2.46	0.03–9.30	1.42
	Feed	21/23 (91.3)	0.81	0.11–2.29	0.60
	Total	45/50 (88.0)	1.70	0.03–9.30	0.86
2015	Grains	35/36 (97.2)	1.68	0.10–6.42	1.33
	Feed	14/15 (93.3)	1.22	0.24–3.14	1.06
	Total	49/51 (96.1)	1.49	0.10–6.42	1.17
2016	Grains	18/18 (100)	0.75	0.10–3.47	0.14
	Feed	8/8 (100)	2.63	0.38–5.72	3.06
	Total	26/26 (100)	1.33	0.10–5.72	0.36
2017	Grains	15/17 (88.2)	1.08	0.28–2.84	0.82
	Feed	8/9 (88.9)	0.69	0.12–1.15	0.65
	Total	23/26 (88.5)	0.95	0.12–2.84	0.75
2018	Grains	46/50 (92.0)	0.48	0.01–4.15	0.20
	Feed	9/10 (90.0)	0.37	0.02–1.06	0.27
	Total	55/60 (91.7)	0.46	0.01–4.15	0.17
2019	Grains	9/32 (28.1)	0.97	0.10–3.44	0.20
	Feed	14/14 (100)	0.51	0.10–1.02	0.40
	Total	23/46 (50.0)	0.69	0.10–3.44	0.40
2020	Grains	30/34 (88.2)	1.46	0.09–3.22	1.48
	Feed	6/8 (75.0)	1.31	0.10–2.58	1.23
	Total	36/42 (85.7)	1.44	0.09–3.22	1.35
2021	Grains	27/30 (90.0)	1.50	0.04–22.23	0.50
	Feed	5/5 (100)	2.34	0.40–9.64	0.53
	Total	32/35 (91.4)	1.62	0.04–22.23	0.53

**Table 2 toxins-14-00444-t002:** Recent data on occurrence of fumonisins in grains and feed in Croatia and other European countries (non-inclusive lists of research of different groups of authors).

Year **	No/Type of Samples	Positive	FBs	Mean (Median) (mg/kg) *	Range (mg/kg) *	Reference
**Croatia**						
1996	105 maize	97%	FB1 FB2	0.65	0.01–11.66	[45]
1997	104 maize	93%	FB1+ FB2	0.13	0.01–2.52	
2002	49 maize	100% 13.3%	FB1 FB2	0.46 1.09	0.14–1.38 0.07–3.08	[27]
2007	37 cereals & feed	27%	FB1+ FB2 + FB3	3.69	0.20–20.70	[43]
2012	40 maize	100%	FBs	1.13 and 0.95	0.03–5.88	[31]
2012	30 pig feed	96%	FBs	0.41	0.03–1.04	[30]
2012	46 maize	67.4%	FBs	-	0.03–25.20	[32]
**Other**	**countries**					
2004–2007	26 maize (FR)	100%	FB1 FB2 FB3	1.43(1.21) 0.48(0.39) 0.25(0.23)	0.35–3.81 0.10–1.23 0.07–0.71	[40]
	56 maize (AR)	96.4% 87.5% 85.7%	FB1 FB2 FB3	0.42(0.29) 0.12(0,06) 0.07(0.05)	<0.01–2.59 <0.01–0.81 <0.01–3.40	
2005–2009	Grains 43 (ELISA)	77%	FB1+ FB2	1.41(1.41)	0.03–7.71	[41]
	Grains & feed 46 (HPLC) (PT, ES, IT, GR, CY)	57%	FB1+ FB2	6.26 (5.30)	0.37–36.39	
2006 2007 2008	697 (IT) 742 (IT) 819 (IT)	high	FB1+ FB2+ FB3	10.9 6.0 4.8	<LOQ-77.0 <LOQ-26.3 0.1–19.0	[42]
2009–2010	14 grains, incl. maize, (Central Europe including HR)	36%	FB1+ FB2	0.93 (0.27)	0.04–7.68	[46]
2012 2013 2014 2015	51 maize (RS) 51 maize (RS) 51 maize (RS) 51 maize (RS)	100% 100% 100% 100%	FB1+ FB2+ FB3	5.66(4.44) 6.24(4.25) 8.31(4.86) 2.73(2.23)	0.31–17.81 0.14–21.51 0.39–34.36 0.28–7.88	[44]
2014–2016	433 grains (oats, wheat, barley, triticale, rye, spelt) (SI)	ND	FB1+ FB2	ND	ND	[47]
2016	91 maize (BE)	25%	FB1+ FB2+ FB3	0.001(0)	0.05–0.07 0.03–1.78 0.03–6.29	[48]
2017	81 maize (BE)	19.4%		0.07(0)		
2018	85 maize (BE)	61.2%		0.33(0.06)		
2013–2015	115 maize, 84 wheat, 58 barley (BiH)	67% 50% 26%	FBs	1.26 0.41 0.15	0.04–3.28 0.04–0.33 0.04–0.23	[49]
2015–2019	98 maize (ES)	70% 55%	FB1 FB2 FB1 + FB2	4.7(1.08) 1.32 (3.74) 5.73 (1.31)	50.1–260 12.1–122 63.1–260	[50]

* Mean/median/range values are included if available and presented in mg/kg irrespective of the units used in the published texts; ** Year of sampling, if available; ND—not detected, PL—Poland, ES—Spain, BiH—Bosnia and Herzegovina, BE—Belgium, SI—Slovenia, HR—Croatia, PT—Portugal, IT—Italy, GR—Greece, CY—Cyprus. NE—northern Europe, SE—southern Europe, EE—eastern Europe.

## Data Availability

Not applicable.

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
