# Peer review of "Determination of Fumonisins in Grains and Poultry Feedstuffs in Croatia: A 16-Year Study"

_toxins, 2022, doi:10.3390/toxins14070444_

Round 1

Reviewer 1 Report

The aim of this research was to evaluate the occurrence of FBs (FB1+FB2 and FB3) in maize samples and poultry feed during a sixteen-year period. For that purpose, we have analysed samples with the commercial enzyme-linked immunosorbent assay (ELISA) and confirmed all suspect and positive samples with high-performance liquid chromatography method with fluorescence detection (HPLC-FLD). 

The abstract is very general and needs redescription.

In material and method is description of samples very shortly. Probably is necessary to do in supplementary file, resources and location of samples, not only percentual description. Properties of each samles can influence mycology.

Results are described very shortly and need more information. For example, some samples are production of mycotoxins related by location or time of collection and also some differences of samples.

Table 2. Recent data on occurrence of fumonisins in grains and feed in Croatia and other European 270 countries (non-inclusive lists of research of different groups of authors) in discussion is little be confusing. Manuscript is research with results and can be described, but in conclusion is discussion written very well.

Conclusion is very general and needs evaluation of interesting results.

The supplementary file is in figure godina/ year, please correct. This part can be only in supplementary file, not in manuscript. Please reject the Croatian language from figures in the supplementary file.

Author Response

Responses to Reviewer #1

We really appreciate all the comments and suggestions. Please, find below the responses to your comments.

Comment no.1.

The aim of this research was to evaluate the occurrence of FBs (FB1+FB2 and FB3) in maize samples and poultry feed during a sixteen-year period. For that purpose, we have analysed samples with the commercial enzyme-linked immunosorbent assay (ELISA) and confirmed all suspect and positive samples with high-performance liquid chromatography method with fluorescence detection (HPLC-FLD).

The abstract is very general and needs redescription

Answer

We have tried to prepare the abstract according to the guidelines of the Journal that states: "The abstract should be a total of about 200 words maximum", and thus unintentionally made it too general. Thank You for your observation. We have changed the abstract to make it more specific. All the changes in the Revised Manuscript are written in red. Please find the changed abstract here as well.

Abstract: Fumonisins are a group of closely related mycotoxins produced by Fusarium, Alternaria alternata and Aspergillus species. Their occurrence is correlated with various factors during growth, processing and storage. Fumonisins occurrence data in the literature mainly include the B group of fumonisins (FB1 & FB2) in raw materials, showing high frequency of positive samples in a wide range of concentrations. In this study, a total of 933 grains (63.7%) and poultry feed (36.3%) samples, collected in the 16-year period (2006-2021), were analysed with commercial enzyme-linked-immunosorbent assay for detection of three fumonisins (FB1, FB2 & FB3). All positive and suspect samples were confirmed with high-performance-liquid-chromatography method with fluorescence detection. Overall, we have determined high occurrence of FBs in grains and poultry feed in all tested years, while the lowest occurrence was determined in 2019, followed by 2009 and 2008. Although, contamination levels varied from year-to-year, majority of analyzed samples in all tested years were around 1 mg/kg, while the maximum values varied from 3 mg/kg to 22.23mg/kg. This study highlights the importance of regular monitoring of raw materials and understanding of the fate of FBs in the food chain in order to avoid undesirable health effects in animals and accompanied economic losses.

Comment no.2.

In material and method is description of samples very shortly. Probably is necessary to do in supplementary file, resources and location of samples, not only percentual description. Properties of each samples can influence mycology.

Answer

Thank you for the comment. Unfortunately we do not have specific information regarding origin and conditions of production, storage, transport, etc. But we have added additional information about type and location of samples in Material and Method section and in Supplementary materials (Tables S1, S2 and Figure S3).

Comment no.3.

Results are described very shortly and need more information. For example, some samples are production of mycotoxins related by location or time of collection and also some differences of samples.

Answer

Thank you for the comment. As stated in previous comments, additional information about samples in Material and Method section, Results section and in Supplementary materials.

Comment no.4.

Table 2. Recent data on occurrence of fumonisins in grains and feed in Croatia and other European 270 countries (non-inclusive lists of research of different groups of authors) in discussion is little be confusing. Manuscript is research with results and can be described, but in conclusion is discussion written very well.

Answer

Thank you for the comment. We have added a paragraph in the discussion section that makes inclusion of the Table in this text more relevant. Here is the added information.

It is clear from the data that FBs have probably been present in different commodities in Croatia and in other countries in this climatic region, but the onset of awareness of their potential harmful effects and availability of more sensitive analytical methods helped in defining the risk of these mycotoxins. However, well-known climatic changes are probably the most important contributing factor for the incidence of higher maximum concentration levels noticed in the more recent surveys.

Comment no.5.

Conclusion is very general and needs evaluation of interesting results.

Answer

Thank you for the comment. We have changed the conclusion so that it includes evaluation of our results. Here is the added information.

  1. Conclusions

Contamination of grains and feed with Fusarium species is a significant mycotoxicological risk due to the often contamination of these moulds and their toxigenic potential.

  • In our study, we have applied optimized and validated analytical methods (ELISA and HPLC-FLD) for the quantification of three B-group fumonisins (FB1, FB2 & FB3) in 933 samples (maize and wheat grains, soybean seed and compound poultry feed) in the 16-year period (2006-2021). The samples originated from different facilities settled in the lowland of northern part of Croatia.
  • The results of this survey indicate that FBs occur frequently in grains and poultry feed marketed in this part of Croatia, and that FBs require the attention of the feed industry and poultry producers. The lowest occurrence of FBs in grains and poultry feed was determined in 2019, followed by 2009 and 2008.
  • Contamination levels of the majority of analyzed samples in all tested years were around 1 mg/kg, while the maximum values varied from 3 mg/kg to 22.23mg/kg. Detected variations of incidence and contamination levels of FBs in analysed samples were probably associated with different contributing factors that favour growth and toxigenicity of Fusarium moulds during plants growth and/or storage.
  • Such high occurrence of FBs (even at concentrations below the maximum tolerable levels) might lead to long term exposure of animals to FBs leading to undesirable health effects and significant economic losses. For the Risk Assessment of potential harmful effects of FBs on animal health, it is important to include all the data about production, transport and storage of raw materials and compound feed.
  • Since environmental factors are important for infection of grains with Fusarium species and production of FBs, prevention of mycotoxin contamination in the field and later during transport and storage is the necessary step in management and control of contamination of raw materials and compound feed. Constant monitoring using available rapid and accurate methods in grains and feed will aid in compliance with current regulations and production of safe high quality feed.

Comment no.6.

The supplementary file is in figure godina/ year, please correct. This part can be only in supplementary file, not in manuscript. Please reject the Croatian language from figures in the supplementary file.

Answer

Thank you for the comment. We used the original Figure from the Document and that is why we did not change it. However, we have now adapted the Figure according to your suggestion and removed Croatian language from figures in the supplementary file. We have also removed supplementary file data from the main manuscript.

Reviewer 2 Report

The introduction is very complete and the state of the art, however, some parts can be moved into the discussion section, such as studies reported in lines 97-129.

In both the abstract and sampling sections, the number of poultry feed samples analyzed is not clear. In addition, detailed information on samples must be provided, for example, feed composition and ingredients.

Regarding the Fumonisin determination by HPLC and fluorescence detection, LOD and LOQ are provided, however, no information regarding recovery, matrix effects, etc are provided. Validation parameters must be included.

Line 108: are those contents in the same range as the above-mentioned ones?

Line 117: contents must be provided, as for the other studies

Line 141: from 2012 to 2016 or 2012-2016 (instead of all the years)

In tables and figures must be indicated in a clear manner that the results correspond to the sum of the 3 FB (FB1+FB2+FB3)

Why in table 1 2006 is bold?

In table 1 it would be better to indicate the number of positive samples of total samples and the percentage in ()  à positive samples/total samples (%)

Line 202: imported instead of importrd

Line 205: I think the authors want to say “are” originated

Why some references along the manuscript are written in red color?

Line 229: I don’t understand why the authors say “respectively” because there are 4 species but 2 content values

Line 265: In table 2 instead of in “a” table 2

Why some values in table 2 are written in red?

Lines 286 and 288: What do the authors think could be the reason for these high values?

Line 300: can the authors briefly explain the BIOMIN project?

Line 327: Abbreviate EFSA in order to use it then in the abbreviated form

Line 354: This value is different from the others; it could be a mistake? Are the authors sure about this value? This value is too high compared to the others, thus, the right axis in figure 3 is higher and the results cannot be compared easily

Line 432: Optical Densities (OD) à thus, it can be used abbreviated in line 437

Can the authors provide a chromatogram? Why the Retention Times are so different?

Author Response

Responses to Reviewer #2

We really appreciate all the reviewers' suggestions and valuable comments.

Comment no.1.

The introduction is very complete and the state of the art, however, some parts can be moved into the discussion section, such as studies reported in lines 97-129.

Answer

Thank you for the comment. We have removed the studies in the lines 97-129 in the discussion section and changed this part of introduction:

A number of surveys on fumonisin occurrence in feed in Croatia and other European countries, conducted by different research groups, showed frequent contaminations at various contamination levels. Occurrence studies on FBs are mainly focused on the presence of FB1 and FB2 in raw materials and only small scale of studies included FB3 and their presence in poultry feed [10,23-32].

Comment no.2.

In both the abstract and sampling sections, the number of poultry feed samples analyzed is not clear. In addition, detailed information on samples must be provided, for example, feed composition and ingredients.

Answer

Thank you for the comment. We have added additional information about samples in Material and Method section, Results section, as well as in in Supplementary Materials (Tables S1, S2 and Figure S3). Unfortunately, we do not have detailed information about the feed composition of the each sample that was analysed. The samples were taken in a scope of producer's monitoring of ingredients and compound feed. We have added type of samples and their numbers to make data more precise. Please find the changed text here as well:

A total of 933 samples of grains (maize grains, wheat grains and soybean seeds) and poultry feed (feed for broilers, laying hens and uncategorized poultry feed) were collected during the period from 2006-2021 from several storage facilities situated in the lowland of northern Croatia (Supplementary Materials, Figures S4, S5). Croatia is situated in southeast Europe adjoining the Mediterranean, central and southeast Europe. It lies between latitude 42° 23’ and 46° 33’ north, and between longitude 13° 30’ and 19° 27’ east. Geographical coordinates of this sampling area are following north (46° 33' N, 16° 22' E), west (45° 28' N, 14° 54' E), south (44° 50' N, 15° 21' E), and east (45° 44' N, 18° 26' E). All samples were taken regularly at the time of filling and in the presence of authorized doctor of veterinary medicine. The samples derived from feed factories that included monitoring of mycotoxins in their control and management programs. The only available data on raw material origin was that it was marketed in this part of Croatia.

Comment no.3.

Regarding the Fumonisin determination by HPLC and fluorescence detection, LOD and LOQ are provided, however, no information regarding recovery, matrix effects, etc are provided. Validation parameters must be included.

Answer

Thank you for the comment. We have added information regarding recovery, matrix effect and additional validation parameters. Here is the added information.

The results of evaluated performance criteria for each tested matrix indicated that the methods are fit-for-purpose. In short, the LOQ were 0.25 mg/kg for ELISA test and 0.01 mg/kg for HPLC-FLD method. Linearity was evaluated with each testing. The results were considered accurate if the linearity coefficient of the calibration curve was ≥ 0.990 for both methods. SD values of evaluation of the parameter Trueness for in all tests (different spiking levels and different matrices) were below 30.00. Precision parameters in repeatability (RSDr) and reproducibility conditions (RSDR) were below ≤ 30% for contamination level ≤0.5 mg/kg and below 20% at the contamination level >0.5 mg/kg. Precision parameters in reproducibility conditions (RSDR) were below ≤ 60% for contamination level ≤0.5 mg/kg and below 30% at the contamination level >0.5 mg/kg. Specificity for spiked and blank samples in all tests was 100%. Testing of samples in duplicates showed that results were reproducible and that tested FBs were identifiable and accurate. In HPLC-FLD, method there for no interfering peaks at LOQ for each of the FBs in each tested matrix. The mean recoveries for total (FB1+FB2+FB3) were 93% for ELISA tests and 97% for HPLC-FLD method. Mean coefficients of variation were 10.5 for ELISA and 8.4% for HPLC-FLD method.

Comment no.4.

Line 108: are those contents in the same range as the abovementioned ones?

Answer

Thank you for this comment. If we have understood the question correctly, contamination after three months storage was in a different range. We have modified the sentence in text. Please find here the changed text:

Contamination with FB1 in the pre-harvest maize samples was in the range from 0.20 mg/kg to 5.90 mg/kg and for FB2 from 0.09 mg/kg to 0.74 mg/kg. In stored maize samples, FB1 concentrations were in the range from 0.18 mg/kg to 0.20 mg/kg, while FB2 was not detected

Comment no.5.

Line 117: contents must be provided, as for the other studies

Answer

Thank you for this comment. We have changed this content in the document. Please find the changes here as well.

In 2002, analysis of maize grain samples showed incidence of FB1 of 100%, while FB2 was present only in 6% of analysed samples. Mean concentration of FB1 was 0.46 mg/kg and the range was from 0.14 mg/kg to 1.38 mg/kg. FB2 concentrations in three positive samples were 0.07 mg/kg, 0.11 mg/kg and 3.08 mg/kg.

Comment no.6.

Line 141: from 2012 to 2016 or 2012-2016 (instead of all the years)

Answer

Thank you for this comment. We have made the change in the document according to your suggestion. Here is also the changed text:

The highest occurrence of FBs in grains (over 90% of analyzed samples) was in the years 2012-2016, 2018 and 2021 and lowest in the years 2008 and 2019 (Figure 1).

Comment no.7.

In tables and figures must be indicated in a clear manner that the results correspond to the sum of the 3 FB (FB1+FB2+FB3)

Answer

Thank you for the comment. We have made suggested changes in the Tables and Figures, so that it is clear that the results correspond to the sum of the three FB (FB1+FB2+FB3).

Comment no.8.

Why in table 1 2006 is bold?

Answer

Thank you for this observation. It was left bold by mistake. We have changed it in the document.

Comment no.9.

In table 1 it would be better to indicate the number of positive samples of total samples and the percentage in () à positive samples/total samples (%)

Answer

Thank you for this comment. We have again checked all the data and noticed typing errors in the Table 1. We have changed that (indicated in yellow) and changed the data and columns according to your suggestion.

Comment no.10.

Line 202: imported instead of importrd

Answer

Thank you for this observation. We have corrected the typing error.

Comment no.11.

Line 205: I think the authors want to say “are” originated

Answer

Thank you for the comment. We have change the sentence and the expression to make it more clear. Please find the changes here as well:

Furthermore, the samples collected from the storage facilities were acquired from different sources. Therefore, we were not able to collect information on other important factors such as: plant genetics (resistant or not), environmental conditions in growing area, temperature (maximum daytime temperatures, minimum night-time temperatures, precipitation, atmospheric CO2 levels, humidity, soil nutrient and moisture content, pH, insect damages, presence of other microorganisms, agricultural practices, transport and storage conditions.

Comment no.12.

Why some references along the manuscript are written in red color?

Answer

Thank you for this observation. The red colour was made by mistake. We have changed it in the document.

Comment no.13.

Line 229: I don’t understand why the authors say “respectively” because there are 4 species but 2 content values

Answer

Thank you for this comment. The term "respectively" was supposed to relate to value of 40% for finished feed samples and value of 100% for other commodities. We have left this in the text.

Comment no.14.

Line 265: In table 2 instead of in “a” table 2

Answer

Thank you for this observation. We have changed it in the document.

Comment no.15.

Why some values in table 2 are written in red?

Answer

Thank you for this observation. It was left in red colour by mistake. We have changed it in the document.

Comment no.16.

Lines 286 and 288: What do the authors think could be the reason for these high values?

Answer

Thank you for this comment. Please find the answer below. We have added this sentence also in the manuscript.

The reason for these high values might be explained by various contributing factors such as susceptibility of grains to fumonisin contamination, pre- and post-harvest agricultural practices, differences in climatic conditions during plant growth (high temperatures and low precipitation around silking) and conditions during storage.

Comment no.17.

Line 300: can the authors briefly explain the BIOMIN project?

Answer

Thank you for this comment. We have added a paragraph including brief description of the BIOMIN project. Please find this paragraph here as well.

The BIOMIN Mycotoxin Survey project started in 2004 with the aim to provide the most comprehensive data set on mycotoxin occurrence in the agricultural commodities used for production of feed. They have analysed a great number of samples worldwide for the presence of six mycotoxins (aflatoxin B1, fumonisin, zearalenone, deoxynivalenol, ochratoxin A and T-2 toxin). Besides valuable data on worldwide occurrence of these mycotoxins, the data were also evaluated in respect to different contributing factors (including climate conditions) to provide information of the most important contributing factors for mycotoxin production and identification of the potential risk of mycotoxin on animal health.

Comment no.18.

Line 327: Abbreviate EFSA in order to use it then in the abbreviated form

Answer

Thank you for this comment. We have corrected the text according to your suggestion.

Comment no.19.

Line 354: This value is different from the others; it could be a mistake? Are the authors sure about this value? This value is too high compared to the others, thus, the right axis in figure 3 is higher and the results cannot be compared easily

Answer

Thank you for this comment. We are aware of this high value, and this sample was tested several times with both methods for confirmation. Therefore, we have decided to include it in the data set. The value remained the manuscript. If the reviewer thinks that, we should present data without this value and to point out in the text on this specific sample, we can make the change. However, we are completely sure that the value is correct.

Comment no.20.

Line 432: Optical Densities (OD) à thus, it can be used abbreviated in line 437

Answer

Thank you for this comment. We have changed the text according to your suggestion.

Comment no.21.

Can the authors provide a chromatogram? Why the Retention Times are so different?

Answer

Thank you for this comment. We did not include the chromatogram because we focused more on the results of the incidence and contamination levels in analysed samples. We have provided additional information regarding validation that also includes information about Retention Times (adapted and optimised method included slightly modified separation protocol and solvents that showed the best performance in our validation study). We have added additional sentence when describing retention times:

The method has been adopted, optimized and validated before routine use according to aforementioned method performance criteria.

AND

(retention times using selected separation protocol and solvents were 7, 21 and 24 min respectively)

Reviewer 3 Report

I can confirm that the subject matter of this study (Determination of Fumonisins in Grains and Poultry Feedstuffs in Croatia: A 16-Year Study) is of interest and relevance for publication in Toxins.

Dear Authors, I read with your interesting manuscript. Here are some revisions to improve it.

-        ln 19: Keywords: delate ‘‘FB’ ; grains’ change to: ‘cereals’

-        ln  393: Materials and Methods: total of 933 samples of grains (maize 90%, wheat 6%, soybean 4%) and poultry – maize and wheat - grains, but soybean - seeds will be better

-        ln 395 – add geographical coordinates

-        ln 270: Table 2 – is ‘cereal’ and ‘grains’, I'm a little confused; and what species of cereals in 2007?

-        ln 378: the conclusion should not be a summary of discussion. Make sure the conclusion is short and solid. An idea may be to synthetize in 3-5 bullet the key results of the study, evidences and recommendation. This improvement will increase clearness and readability. Add a practical implications statement.

Author Response

Responses to Reviewer #3

We really appreciate all the reviewers' suggestions and valuable comments.

Comment no.1.

I can confirm that the subject matter of this study (Determination of Fumonisins in Grains and Poultry Feedstuffs in Croatia: A 16-Year Study) is of interest and relevance for publication in Toxins. Dear Authors, I read with your interesting manuscript. Here are some revisions to improve it.

Answer

Thank you for this comment. We appreciate your opinion and suggestions for improvement of our manuscript.

Comment no.2.

ln 19: Keywords: delate ‘‘FB’ ; grains’ change to: ‘cereals’

Answer

Thank you for this comment. We have changed the keywords according to your suggestion.

Comment no.3.

ln 393: Materials and Methods: total of 933 samples of grains (maize 90%, wheat 6%, soybean 4%) and poultry – maize and wheat - grains, but soybean - seeds will be better

Answer

Thank you for this comment. We have changed the text in the manuscript. Please find the changed text here as well:

A total of 933 samples of grains (maize grains, wheat grains and soybean seeds) and poultry feed (feed for broilers, laying hens and uncategorized poultry feed) were collected during the period from 2006-2021 from several storage facilities situated in the lowland of northern Croatia (Supplementary Materials, Figures S4, S5). Croatia is situated in southeast Europe adjoining the Mediterranean, central and southeast Europe. It lies between latitude 42° 23’ and 46° 33’ north, and between longitude 13° 30’ and 19° 27’ east. Geographical coordinates of this sampling area are following north (46° 33' N, 16° 22' E), west (45° 28' N, 14° 54' E), south (44° 50' N, 15° 21' E), and east (45° 44' N, 18° 26' E). All samples were taken regularly at the time of filling and in the presence of authorized doctor of veterinary medicine. The samples derived from feed factories that included monitoring of mycotoxins in their control and management programs. The only available data on raw material origin was that it was marketed in this part of Croatia.

Comment no.4.

ln 395 – add geographical coordinates

Answer

Thank you for this comment. We have added geographical coordinates as you have suggested. This is also included in Material and Methods section and in Supplementary material Figure S5. It is also included here:

Figure S5: Information about geographical origin of samples (adapted from the official Croatian Metrological website with data on evaluation of climate conditions in Croatia (www.meteo.hr). Croatia is situated in southeast Europe adjoining the Mediterranean, central and southeast Europe. It lies between latitude 42° 23’ and 46° 33’ north, and between longitude 13° 30’ and 19° 27’ east. Geographical coordinates of this sampling area are following north (46° 33' N, 16° 22' E), west (45° 28' N, 14° 54' E), south (44° 50' N, 15° 21' E), and east (45° 44' N, 18° 26' E).

Comment no.5.

ln 270: Table 2 – is ‘cereal’ and ‘grains’, I'm a little confused; and what species of cereals in 2007?

Answer

Thank you for this comment. We have added additional information in the Material and Methods and Results sections and in Supplementary materials Figures S3 and S4.

Comment no.6.

ln 378: the conclusion should not be a summary of discussion. Make sure the conclusion is short and solid. An idea may be to synthetize in 3-5 bullet the key results of the study, evidences and recommendation. This improvement will increase clearness and readability. Add a practical implications statement.

Answer

Thank you for the comment. We have changed the conclusion so that it includes evaluation of our results. Here is the added information.

  1. Conclusions

Contamination of grains and feed with Fusarium species is a significant mycotoxicological risk due to the often contamination of these moulds and their toxigenic potential.

  • In our study, we have applied optimized and validated analytical methods (ELISA and HPLC-FLD) for the quantification of three B-group fumonisins (FB1, FB2 & FB3) in 933 samples (maize and wheat grains, soybean seed and compound poultry feed) in the 16-year period (2006-2021). The samples originated from different facilities settled in the lowland of northern part of Croatia.
  • The results of this survey indicate that FBs occur frequently in grains and poultry feed marketed in this part of Croatia, and that FBs require the attention of the feed industry and poultry producers. The lowest occurrence of FBs in grains and poultry feed was determined in 2019, followed by 2009 and 2008.
  • Contamination levels of the majority of analyzed samples in all tested years were around 1 mg/kg, while the maximum values varied from 3 mg/kg to 22.23mg/kg. Detected variations of incidence and contamination levels of FBs in analysed samples were probably associated with different contributing factors that favour growth and toxigenicity of Fusarium moulds during plants growth and/or storage.
  • Such high occurrence of FBs (even at concentrations below the maximum tolerable levels) might lead to long term exposure of animals to FBs leading to undesirable health effects and significant economic losses. For the Risk Assessment of potential harmful effects of FBs on animal health, it is important to include all the data about production, transport and storage of raw materials and compound feed.
  • Since environmental factors are important for infection of grains with Fusarium species and production of FBs, prevention of mycotoxin contamination in the field and later during transport and storage is the necessary step in management and control of contamination of raw materials and compound feed. Constant monitoring using available rapid and accurate methods in grains and feed will aid in compliance with current regulations and production of safe high quality feed.

Round 2

Reviewer 1 Report

The authors corrected all suggestions.

In supplementary fila are numbers with commas in the tables. Correct number with dots.

Author Response

Dear Reviewer,

Thank You for this observation. We have corrected the Tables in Supplementary Materials according to your suggestion.

Kind Regards,

Authors